# The Role of Growth Factors and Signaling Pathways in Ovarian Angiogenesis

**DOI:** 10.3390/cells14191555

**Published:** 2025-10-07

**Authors:** Hanna Jankowska-Ziemak, Magdalena Kulus, Aleksandra Partynska, Jakub Kulus, Krzysztof Piotr Data, Dominika Domagala, Julia Niebora, Aleksandra Gorska, Marta Podralska, Marzenna Podhorska-Okolow, Piotr Chmielewski, Paweł Antosik, Dorota Bukowska, Adam Kaminski, Hanna Piotrowska-Kempisty, Maciej Zabel, Paul Mozdziak, Piotr Dziegiel, Bartosz Kempisty

**Affiliations:** 1Department of Veterinary Surgery, Institute of Veterinary Medicine, Nicolaus Copernicus University in Torun, 87-100 Torun, Poland; magdalena.kulus@umk.pl (M.K.); bartosz.kempisty@umw.edu.pl (B.K.); 2Division of Histology and Embryology, Department of Human Morphology and Embryology, Faculty of Medicine, Wroclaw Medical University, 50-368 Wroclaw, Poland; 3Department of Diagnostics and Clinical Sciences, Institute of Veterinary Medicine, Nicolaus Copernicus University in Torun, 87-100 Torun, Poland; 4Division of Anatomy, Department of Human Morphology and Embryology, Faculty of Medicine, Wroclaw Medical University, 50-358 Wroclaw, Poland; 5Department of Stem Cells and Regenerative Medicine, Institute of Natural Fibres and Medicinal Plants, 60-630 Poznan, Poland; 6Department of Human Morphology and Embryology, Division of Ultrastructure Research, Wroclaw Medical University, 50-358 Wroclaw, Poland; 7Department of Human Biology and Cosmetology, Faculty of Physiotherapy, Wroclaw University of Health and Sport Sciences, Ignacego Jana Paderewskiego 35, 51-612 Wroclaw, Poland; 8Department of Orthopedics and Traumatology, Independent Public Clinical Hospital No. 1, Pomeranian Medical University in Szczecin, Unii Lubelskiej 1, 71-252 Szczecin, Poland; 9Department of Basic and Preclinical Science, Institute of Veterinary Medicine, Nicolaus Copernicus University in Torun, 7 Gagarina St., 87-100 Torun, Poland; 10Department of Toxicology, Poznan University of Medical Sciences, 60-631 Poznan, Poland; 11Division of Anatomy and Histology, The University of Zielona Góra, Licealna 9, 65-417 Zielona Góra, Poland; 12Graduate Physiology Program, North Carolina State University, Raleigh, NC 27695, USA; 13Prestage Department of Poultry Science, North Carolina State University, Raleigh, NC 27695, USA; 14Department of Obstetrics and Gynecology, University Hospital and Masaryk University, 625 00 Brno, Czech Republic

**Keywords:** ovarian angiogenesis, growth factors, signaling pathways

## Abstract

Angiogenesis, the formation of new blood vessels from existing vasculature, is regulated by a balance between pro- and anti-angiogenic factors. In adults, this process typically occurs in response to inflammation, wound healing, and neoplastic growth. Uniquely, the female reproductive system undergoes cyclical and repetitive angiogenesis with folliculogenesis, decidualization, implantation, and embryo development throughout the reproductive cycle. Ovarian angiogenesis involves a coordinated network of signaling pathways and molecular factors. Vascular endothelial growth factor (VEGF) is the primary driver of this process, supported by other regulators such as fibroblast growth factor (FGF) and hypoxia-inducible factor (HIF). Understanding the molecular mechanisms that govern ovarian angiogenesis is essential for developing new diagnostic and therapeutic approaches in reproductive medicine. Vascular dysfunction and impaired angiogenesis are key contributors to various ovarian disorders and infertility, including polycystic ovary syndrome (PCOS). Therefore, in-depth studies of ovarian vascularization are crucial for identifying the pathophysiology of these conditions and guiding the development of effective treatments. Advancing knowledge in this area holds significant potential for innovation in both medicine and biotechnology.

## 1. Introduction

Angiogenesis, the process in which new blood vessels are created (Figure 1), plays a pivotal role in ovarian physiology, particularly during the menstrual cycle, ovulation and corpus luteum formation. The ovary is an organ with an exceptionally dynamic microenvironment, in which angiogenesis is essential to maintain hormonal function and to ensure an adequate supply of oxygen and nutrients for ovarian follicle maturation. Vascular endothelial growth factor (VEGF), fibroblast growth factor (FGF), platelet-derived growth factor (PDGF), and hypoxia-inducible factor (HIF) are examples of angiogenic factors. They play a pivotal role in the regulation of angiogenesis. These factors, by activating appropriate signaling pathways, regulate a number of cellular processes necessary for the formation of new blood vessels such as endothelial cell proliferation or migration. Vascular endothelial growth factor (VEGF) is a principal mediator in the initiation of angiogenesis, as this molecule increases vascular proliferation and permeability, whereas fibroblast growth factor (FGF) participates in the regulation of endothelial cell migration and differentiation. Platelet-derived growth factor (PDGF), conversely, is involved in the stabilization of newly formed vessels, attracting supporting cells such as pericytes. Hypoxia-inducible factor (HIF) is an important regulator of the angiogenic response in hypoxic conditions. It is of particular significance in ovaries, where fluctuations in oxygen levels commonly occur.

The physiological events that occur during the normal estrous cycle are regulated by hormones from the hypothalamus (gonadotropin-releasing hormone; GnRH), the anterior pituitary (follicle-stimulating hormone; FSH and luteinizing hormone; LH), the ovaries (progesterone; P4, estradiol; E2 and inhibins), and the uterus (prostaglandin F2) [1]. These glands function through a system of positive and negative feedback that regulate the entire estrous cycle. The estrous cycle is characterized by great variability among species and even individual variability, as in women, where we observe different cycle lengths (26–35 days) [2]. In all women, there is an increase in FSH during the luteal–follicular transition, which stimulates follicular growth and inhibits B secretion in the early follicular phase. The selection of the dominant follicle of ovulation (DF) occurs during the mid-follicular phase, and the gonadotropin response, insulin-like growth factor (IGF) binding protein expression and degradation, and vascularization have been identified as critical factors in DF selection and progression. Two-thirds of women exhibit two follicular waves, while one-third exhibit three follicular waves per cycle. Women with three waves have longer cycles and a later estradiol and luteinizing hormone (LH) surge. The corpus luteum secretes progesterone, estradiol, and inhibin A in response to LH pulses and peaks 6–7 days after ovulation [3]. As shown in Figure 2, the ovarian follicle consists of an oocyte surrounded by granulosa and theca cells. This represents the basic functional unit of the ovary. The process of follicular growth can be categorized into three distinct phases, distinguished by their developmental stage and gonadotropin dependence: Firstly, primordial, primary, and secondary stages (gonadotropin-independent phase) are characterized by follicular growth. Secondly, there is a transition from preantral to early antral stage (gonadotropin-responsive phase). Thirdly, there is continual growth beyond the early antral stage (gonadotropin-dependent phase), which includes follicle recruitment, selection, and ovulation [4,5,6]. In the gonadotropin-responsive phase, follicle growth is primarily governed by intraovarian regulators (e.g., 61 growth factors, cytokines, and gonadal steroids), thus obviating the need for gonadotropins. Nonetheless, the presence of FSH has been demonstrated to stimulate this process [7,8]. The cells immediately surrounding the oocyte are known as cumulus cells, and these cells accompany the oocyte throughout its development from immature to fully mature ovulatory oocyte. They play a pivotal role in maintaining the oocyte, both in vivo and in vitro [9]. The zona pellucida (ZP) is a glycoprotein interface between oocytes and surrounding cells that covers and protects the mammalian oocyte. The zona pellucida plays an important role in sperm-oocyte interaction, the acrosome reaction, and the blockade of polyspermy [9]. It should be noted that the expression of angiogenic mediators in the ovary is not constitutive but dynamically regulated. Hypoxia accompanying follicle growth and luteinization, gonadotropin stimulation, and inflammatory signals associated with ovulation represent key triggers for the synthesis of VEGF, FGF, PDGF and related factors. By precisely regulating the pathways involved in angiogenesis, it is possible to ensure a dynamic balance between vessel formation and regression, which is essential for the proper functioning of the ovary. Deregulation of these processes has been linked to pathologies such as polycystic ovary syndrome (PCOS) and ovarian cancer. It is important to consider the role of angiogenic factors relative to their function in physiological and pathological conditions.

## 2. Angiogenesis-Promoting Growth Factors

### 2.1. HIF (Hypoxia-Inducible Factor)

Cellular hypoxia is observed when oxygen supply is reduced, leading to insufficient ATP production for physiological functions. To protect themselves from the effects of hypoxia, eukaryotic cells adopt oxygen-sensitive transcriptional responses, such as those triggered by hypoxia-inducible factor (HIF), which includes several heterodimers such as HIF-1, HIF-2, and HIF-3 [10]. Hypoxia-inducible factor-1 (HIF-1), a transcription factor found in multicellular organisms, plays a key role in regulating gene expression under hypoxic conditions, enabling adaptation to a hypoxic environment [11,12,13]. HIF-1 was first described when a hypoxia response element (HRE, 5-RCGTG-3) was isolated from the 3′ terminus of the erythropoietin (EPO) gene. The EPO gene encodes erythrocyte proliferation-stimulating hormone (EPO) and is expressed under hypoxic conditions [14,15]. The process of endothelial cell adaptation is orchestrated by HIF-1 and HIF-2. These cells have been observed to participate in the activation of gene signalling cascades that promote endothelial migration, growth, and differentiation. Initially, HIF-1 is the primary regulator of adaptation during the acute phase of hypoxia. Subsequently, HIF-2 becomes active, resulting in a transient switch between the two HIF proteins. The process of cell death, or apoptosis, is initiated when HIF-1 levels are unable to be reduced during periods of prolonged hypoxia [16,17]. HIF-1 is a heterodimeric transcription factor composed of two subunits: HIF-1α (hypoxia-inducible) and HIF-1β (constitutively expressed), also known as aryl hydrocarbon nuclear translocator (ARNT). These two proteins contain a basic helix-loop-helix motif (bHLH) and a Per-ARNT-Sim (PAS) domain [18]. Posttranslational modifications of the HIF-1α protein have the greatest impact on HIF-1 activity [19]. Three isoforms of HIF-α have been described: HIF-1α, HIF-2α, and HIF-3α, which share common functional domains in their structure [20]. HIF-1 is expressed in a wide range of human and mouse tissues and plays a role in numerous physiological responses to hypoxia [12]. It has been demonstrated that there are discrepancies in the expression profiles and target sites of HIF isoforms. HIF-1α is expressed in most tissues. During embryonic development, HIF-2α is most abundant in vascular endothelial cells, whereas in the postnatal period, it is expressed in specific cell populations, including renal fibroblasts and liver hepatocytes, and in almost all transformed cell lines [18,21]. HIF-1 has a short half-life and is strongly regulated by oxygen [18]. Under normoxic conditions, HIF-1 protein expression remains minimal and typically falls below detection thresholds. During hypoxia, HIF-1α stabilizes and translocates to the nucleus, where it forms dimers with HIF-1β. The resulting HIF complex is activated. The HIF complex then binds to HREs in the regulatory regions of target genes and binds transcriptional coactivators, inducing gene expression [21,22]. The ovarian follicle is a highly specialized structure that requires oxygen for metabolic activity during development. This is achieved by creating a low-oxygen environment for the maturing oocyte within it. During follicular development and corpus luteum formation, hypoxia plays an important role in facilitating oxygen regulation. Gonadotropic signals via FSH and LH are involved in maintaining HIF activity, upregulating VEGF, and inducing angiogenesis. Experimental evidence derived from the in vitro culture of rat granulosa cells indicates that hypoxia-inducible factor (HIF) is activated by gonadotropins, including follicle-stimulating hormone (FSH) [23]. Studies performed on luteinized human granulosa cells treated with human chorionic gonadotropin (hCG) and cultured under hypoxic conditions (1% and 5% oxygen) demonstrated increased expression of HIF1A and HIF2A [24]. Culture of bovine granulosa cells under low-oxygen conditions upregulates angiogenesis-related genes. These genes are known as markers of early luteinization [24]. The multiplicity of HIF-1 regulatory mechanisms offers a plethora of potential avenues for therapeutic intervention.

### 2.2. Transforming Growth Factor-Beta (TGF-β)

Transforming growth factor-β (TGF-β) is a signaling molecule that is characterized by a high degree of versatility, in that its biological effects are context-dependent. It is a precise regulator of tissue homeostasis, responsible for promoting or inhibiting cell proliferation depending on the cellular and microenvironmental conditions. TGF-β is widely described as capable of driving fibroblast transformation in cooperation with epidermal growth factor (EGF) [25]. It is considered an important regulator of ovarian physiology, particularly through its influence on vascular remodeling during follicular growth, ovulation, and corpus luteum formation [26]. TGF-β exerts its effects by binding to TGF-β type I and type II receptors, which are serine-threonine kinase receptors. Following the binding of TGF-β to the receptor, phosphorylation of type I receptors occurs, resulting in the activation of intracellular proteins belonging to the SMAD family (SMAD2 and SMAD3). Upon activation, SMAD2/3 bind to coactivators such as SMAD4 and translocate to the nucleus, where they affect the transcription of target genes [27]. In addition to the SMAD-dependent pathway, TGF-β also activates SMAD-independent pathways, including the mitogen-activated protein kinase (MAPK), phosphatidylinositol 3-kinase/protein kinase B (PI3K/AKT), and Rho-GTPase pathways, which are also involved in the regulation of angiogenesis, as showed at Figure 3. TGF-β promotes the increased production of vascular endothelial growth factor (VEGF), which is a key regulator of endothelial cell proliferation, migration and permeability [28]. By modulating angiopoietins, it promotes vascular stabilization through its effects on Ang-1, which enhances endothelial–pericyte interactions through activation of the Tie-2 receptor, and Ang-2, which destabilizes vessels, facilitating sprouting [29]. In addition to these well-characterized mediators, TGF-β is involved in regulating endothelin-1 in the ovaries. It promotes endothelial proliferation and remodeling via ETA/ETB receptors, as well as matrix metalloproteinases (MMPs). MMPs degrade extracellular matrix components, enabling the expansion of neovascularization [30,31]. It is important to emphasize that the effects of TGF-β are not limited to proangiogenic activity alone. By inducing anti-angiogenic proteins, including thrombospondin-1 (TSP-1), TGF-β influences the temporal and spatial inhibition of endothelial proliferation, thereby preventing excessive vascular growth [32]. This functional duality is evident throughout the ovarian cycle. In the follicular phase, TGF-β and other factors support angiogenesis, which is essential for follicle maturation. In the luteal phase, TGF-β drives rapid neovascularization, which is essential for corpus luteum formation and progesterone synthesis [33]. Aberrant regulation of TGF-β signaling can have pathological implications. In ovarian cancer, for example, TGF-β promotes tumor angiogenesis, driving growth and metastatic spread. In polycystic ovary syndrome (PCOS), meanwhile, it may be one of the causative factors leading to cyst development [34]. Therefore, the TGF-β pathway has both physiological necessity and pathological risk, making it a promising target for treating ovarian dysfunction.

### 2.3. Hepatocyte Growth Factor (HGF)

Hepatocyte growth factor (HGF) is a multifunctional cytokine that regulates key processes involved in angiogenesis, including cell proliferation, migration, and differentiation. In response to various environmental stimuli, HGF binds to and activates the MET tyrosine kinase receptor, thereby triggering a series of signaling pathways that support both cell growth and angiogenesis [35]. In a manner analogous to other growth factors, HGF exerts an effect on VEGF, thereby supporting and enhancing its angiogenic action, as is also the case with other growth factors. It has been demonstrated that HGF activates signaling pathways which are responsible for the growth and survival of endothelial cells. The pathways in question include PI3K/AKT and MAPK/ERK. Furthermore, HGF indirectly stimulates angiogenesis by regulating the proteolysis of the extracellular matrix (ECM), thereby facilitating the migration of endothelial cells to sites of new vessel formation [36]. It is evident that HGF performs a pivotal role in the process of regeneration, with particular reference to the process of tissue repair following injury. The ability of the substance to stimulate angiogenesis has been demonstrated to facilitate the delivery of oxygen and nutrients to damaged areas. Furthermore, HGF has been demonstrated to possess anti-inflammatory properties that promote angiogenesis in conditions of chronic inflammation, such as cardiovascular disease or during myocardial regeneration after myocardial infarction [37]. In the context of pathology, hepatocyte growth factor (HGF) and its receptor, MET, are often overexpressed in a number of cancers, leading to excessive angiogenesis and providing support for the growth and invasiveness of cancer cells. The activation of the HGF/MET pathway in cancer cells facilitates the formation of a network of vessels supplying nutrients, thereby enabling metastasis to distant organs. Consequently, HGF has emerged as a pivotal target in antiangiogenic cancer therapy, and HGF/MET pathway inhibitors are being extensively investigated for their therapeutic potential [38]. In summary, HGF plays a pivotal role in angiogenesis through its effects on endothelial cells and its interaction with other angiogenic factors, such as VEGF. The role of HGF in angiogenesis is of particular importance in both physiological processes, such as tissue regeneration, and pathological conditions, where it can promote cancer progression by forming new blood vessels.

### 2.4. Platelet-Derived Growth Factor (PDGF)

The platelet-derived growth factor (PDGF) family consists of proteins with strong mitogenic and chemotactic capabilities, essential for vascular formation as well as tissue development and repair. Within ovarian angiogenesis, PDGF plays a central role in coordinating interactions between endothelial cells and pericytes, a process critical for the stabilization and proper maturation of newly established blood vessels. PDGF proteins exist as either homo- or heterodimers, formed from four distinct polypeptide chains: A, B, C, and D [39]. The effects of these ligands are exerted through two main receptors, PDGFR-α and PDGFR-β, which are activated upon ligand binding. The signaling via PDGFB and PDGFD through PDGFR-β is of particular importance for the proliferation and migration of pericytes and perivascular cells. During ovarian angiogenesis, PDGF contributes to the coordination between endothelial cells and pericytes, a process essential for the proper stabilization and maturation of newly formed vasculature. PDGFs constitute a family of homo- and heterodimers assembled by four distinct polypeptide chains: A, B, C, and D [40]. Two distinct receptors, PDGFR-α and PDGFR-β, are bound and activated by PDGF. Signaling via PDGFB and PDGFD at the receptor PDGFR-β is of critical importance for the proliferation of pericytes and perivascular cells, as well as for their migration along the newly formed vessel [40]. In the ovary, PDGFB, PDGFD, and PDGFR-β are expressed in various locations, including oocytes, theca cells, and the stroma. Human granulosa cells have been shown to express PDGFA, PDGFB, and PDGFR-β [40], while mouse and rat ovarian cells also show the presence of PDGFB and PDGFR- in oocytes and granulosa cells [41,42]. Evidence indicates that PDGF signaling influences the development of theca cells and steroidogenesis in the ovary [42]. Importantly, PDGFB/PDGFR-β activity appears to counteract the action of anti-Müllerian hormone (AMH), participating in early folliculogenesis and promoting the activation of primordial follicles [40,43]. Upon binding to PDGFR-α or PDGFR-β, PDGF induces receptor dimerization and autophosphorylation of intracellular tyrosine residues, initiating signaling cascades, including PI3K/Akt and MAPK/ERK pathways. These pathways facilitate the proliferation and migration of endothelial cells and pericytes, supporting angiogenesis [44]. Dysregulation of PDGF signaling has been linked to ovarian pathologies such as excessive pericytic proliferation contributing to cyst formation and other angiogenic disorders associated with polycystic ovary syndrome (PCOS). In ovarian cancer, increased PDGF/PDGFR activity promotes tumor angiogenesis, supporting tumor growth and progression [44]. Furthermore, increased PDGFR expression has been linked to resistance to therapy and more aggressive tumor behavior.

### 2.5. Matrix Metalloproteinases (MMP-2, MMP-9)

Matrix metalloproteinases (MMPs) constitute a group of proteolytic enzymes that are responsible for the degradation of extracellular matrix (ECM) components. In the process of angiogenesis, matrix metalloproteinases (MMPs) play a pivotal role as they facilitate tissue remodeling, endothelial cell migration and the formation of new blood vessels, particularly in the ovary (Figure 4). Among matrix metalloproteinases, MMP-2 and MMP-9 are of particular importance in the context of ovarian angiogenesis, given their involvement in the degradation of type IV collagen, the principal component of the basement membranes that surround blood vessels [44,45]. MMP-2 is a proteolytic enzyme with the capacity to degrade a range of ECM components, including type I and IV collagen, gelatin, and elastin. In contrast, MMP-9 is unable to directly proteolyze type I collagen. Instead, it digests type IV collagen. The activity of MMP-2 is regulated by MMP-activating enzymes and inhibitors, including tissue inhibitors of metalloproteinases (TIMP-2). MMP-2 is in an inactive state (pro-MMP-2) until proteolytic activation, which is typically initiated by other proteases, with a pivotal role of MT1-MMP (membrane-type 1 MMP) [44]. Pro-MMP-9 requires proteolytic activation which is dependent on enzymes such as plasmin and MT1-MMP [46]. MMP-9 plays a key role in endothelial cell migration and matrix remodeling during angiogenesis, particularly in the dynamic processes occurring in the ovary. The process of ovarian angiogenesis is largely dependent on the degradation and remodeling of ECM, enabling endothelial cells to migrate in order to form new blood vessels. MMP-2 and MMP-9 play an important role in this process, particularly in menstrual cycle, during which alternating growth and regression of blood vessels in the ovarian follicles and corpus luteum are reported [47]. During the follicular phase, matrix metalloproteinases (MMP-2 and MMP-9) participate in ovarian tissue remodeling, therefore facilitating follicular growth and vascularization [48]. Ovulation is a process that is dependent on the activity of matrix metalloproteinases (MMPs) because the rupture of the follicle requires ECM remodeling. The elevated levels of MMP-2 and MMP-9 during this phase facilitate collagen degradation and endothelial cell migration as these two processes are indispensable for the formation of blood vessels in the corpus luteum. In addition, activation of MMPs is supported by an increase in hormones such as LH (luteinizing hormone), which promotes the synthesis and activity of MMPs [49]. After ovulation, MMP-2 and MMP-9 are important for the formation of new vessels in the corpus luteum, as this highly vascularized structure is responsible for progesterone production. The remodeling of the corpus luteum vasculature by MMPs is of great importance for the maintenance of corpus luteum function. Dysregulation of this process can result in embryo implantation complications or menstrual cycle disruptions [50]. The expression and activity of MMP-2 and MMP-9 are subject to precise regulation by a range of signaling factors, i.e., cytokines, hormones, and growth factors (such as VEGF) and FGF. VEGF was demonstrated to stimulate the secretion of MMP-9 by endothelial cells, therefore promoting the degradation of basement membranes and facilitating cell migration [49]. Interleukins such as IL-1β and IL-8 stimulated the expression of MMP-9, particularly in the context of inflammatory processes occurring during ovulation. These cytokines regulate the activity of MMP-9 during angiogenesis by inducing the expression of genes related to matrix degradation [51]. The activity of MMP-2 and MMP-9 is inhibited by tissue inhibitors of metalloproteinases (TIMPs), particularly TIMP-2 and TIMP-1. The equilibrium between MMPs and TIMPs is of paramount importance for the optimal progression of angiogenesis. Prolonged MMP activity can result in pathological angiogenesis, whereas a deficiency of MMPs can impede neovascularization. Imbalanced activity of MMP-2 and MMP-9 can result in pathological conditions within the ovary, including the formation of ovarian cysts, the development of PCOS, and the onset of ovarian cancer. In PCOS, elevated levels of MMP-9 may contribute to uncontrolled angiogenesis that promotes cyst formation. Elevated levels of both MMP-2 and MMP-9 were observed in ovarian cancer. These phenomena were linked to tumor cell invasion and tumor angiogenesis which contribute to tumor growth and metastasis [52].

## 3. Signaling Pathways Involved in Angiogenesis

### 3.1. FGFR

The mammalian FGF signaling pathway consists of eighteen secreted proteins that interact with four FGF receptors (FGFRs). These receptors exhibit intracellular tyrosine kinase activity. During embryonic development and organogenesis, FGFs maintain progenitor cells and mediate their growth, differentiation, survival, and patterning. The interaction between FGF and their receptors is regulated by protein and proteoglycan cofactors and extracellular binding proteins as well. FGFs also mediate interactions with cytosolic adaptor proteins and the intracellular signaling pathways RAS-MAPK, PI3K-AKT, PLC, and STAT [53]. FGFs exert their effects by binding with high affinity to four distinct but closely related FGFRs (FGFR1, -2, -3, and -4). Binding of FGF to the FGFR leads to receptor dimerization and autophosphorylation of tyrosine residues within intracellular domain of the receptor. Autophosphorylation of the FGFR results in binding of signal transducing molecules. This process leads to the activation of signaling pathways and to cellular responses such as proliferation, migration, and differentiation. The Ras pathway is important in the biological responses to FGFs. Key components of this pathway include the mitogen-activated protein kinases Erk1 and Erk2 which become phosphorylated. After translocation into the nucleus they regulate the activity of certain early-response transcription factors such as Myc and Fos. FGF is known to bind to heparan sulphate (HS)/heparin. FGF is protected from degradation by binding to HS on the cell surface and in the extracellular matrix. FGFs can be released from FGF-HS complex by heparinases. FGF can participate in repair following injury [54]. FGF7 is expressed in theca cells, but not in granulosa cells or oocytes [55]. In contrast, FGFR2 is readily detected in granulosa cells, oocytes, cumulus and theca cells [56].

### 3.2. VEGFR

Angiogenesis is strongly regulated by vascular endothelial growth factors (VEGFs), mainly via VEGFR-2, but additional pathways (FGF, PDGF, angiopoietins, Notch, TGF-β) also play important roles in vascular development and pathology. The predominant receptor expressed in vascular endothelial cells, VEGFR-2, plays a crucial role in angiogenesis. The human VEGFR-2 gene (KDR) is located on chromosome 4q12 [57]. VEGFR-2 is predominantly found in vascular endothelial cells and acts as a key signal transducer for angiogenesis through the PLCγ -PKC-MAPK, PLCγ -PKC-eNOS-NO, TSAd-Src-PI3K-Akt, SHB-FAK-paxillin, SHB-PI3K-Akt, and NCK-p38-MAPKAPK2/3 pathways [58]. VEGFR-2 supports endothelial cell survival mainly via the PI3K–Akt pathway, with TSAd–Src interactions facilitating recruitment of signaling intermediates. VEGFR-2 consists of 1356 amino acids, including a signal peptide (119 aas) and a mature protein (201,356 aas). As a mature protein, VEGFR-2 is divided into an extracellular domain (ECD, 20,764 aa), a transmembrane domain (TMD, 765,789 aa), a juxtamembrane domain (JMD, 790,833 aa), and a tyrosine kinase catalytic domain (TKD, 8,341,162 aa), as well as including the ATP binding domain (TKD1, 834–930 aa), the kinase insertion domain (KID, 931–998 aa) and the phosphotransferase domain (TKD2, 999–1162 aa), and the flexible C-terminal domain (CTD, 1163–1356 aa) [58]. The extracellular domain (ECD) comprises seven immunoglobulin-like subdomains: IgD1, IgD2, IgD3, IgD4, IgD5, IgD6, and IgD7. The binding of the receptor to its ligands is regulated by IgD1, while IgD2 and IgD3 are important for binding to dimeric VEGF, which then activates VEGFR-2. IgD2 and IgD3 are essential for VEGF binding; D4–D7 primarily stabilize the receptor and regulate dimer formation, rather than directly controlling ligand binding speed or separation [59]. VEGFR-2 is activated by VEGF-A, VEGF-C, and VEGF-D. Cellular signaling through VEGFR-2 is triggered by ligand dimer binding to the receptor’s Ig-like domains 2 and 3. Next, VEGFR-2 forms homo- and heterodimers, and tyrosine residues in tyrosine kinase domain of intracellular regions and in the carboxyterminal domain become phosphorylated. Numerous signaling molecules bind to VEGFR-2 dimers, triggering additional signaling pathways influencing endothelial cell properties and the vascular environment. VEGFR2 mediated signaling is vital in maintaining the physiological functions of blood vessels [60]. The extracellular domain (ECD) of VEGFR-2 is part of the VEGF/VEGFR-2 system and has Ig-like subdomains, linkers, and glycosylation sites. The Ig-like subdomains and glycosylation sites are very important for the formation of the VEGF/VEGFR-2 system and receptor dimerization (the process by which two receptors bind together) upon ligand binding, as well as the maintenance of monomeric VEGFR-2 in the absence of ligand. Each VEGF monomer has two α-helices and five β-sheets that form a central antiparallel beta sheet [60]. The VEGFR-2 ECD has the ability to bind to ligands, where Ig-like subdomains 2, 3, 4–7 play a significant role in this process. VEGFR-2 also contains 18 N-glycosylation sites that are vital for ligand binding, stabilization, and proangiogenic signaling [60,61,62,63]. VEGFR-2 is activated by dimerization mediated by VEGFs. This dimerization enables the trans-phosphorylation of the tyrosine kinase domain (TKD) of VEGFR-2 and affects cellular processes such as survival, proliferation, migration and blood vessel formation [64]. Important phosphorylation sites in VEGFR-2 include Tyr951 (Y951), Tyr1054 (Y1054), and Tyr1059 (Y1059). These sites have a crucial function in activating the kinase activity [65]. Moreover, autophosphorylation at Tyr1175 (Y1175) and Tyr1214 (Y1214) sites in the carboxyl-terminal domain (CTD) is also vital for VEGFR-2 activation and signaling. Upon activation, VEGFR-2 interacts with several signaling molecules including PLCγ, PI3K, as well as with adaptor proteins such as SHB, Sck, and NCK. VEGFR-2 transmits extracellular signals to the cytoplasm and subsequently activates various downstream signaling pathways. The main signaling pathway, PLCγ -PKC-Raf-MEK-MAPK, transmits signals from VEGFR to the nucleus, activating DNA synthesis and promoting endothelial cell proliferation. Ultimately, this affects the physiological properties of the endothelial cells and the entire vascular environment [60,66].

### 3.3. Notch

The Notch signaling pathway is a key regulator of ovarian physiology, acting during embryonic development as well as after birth by controlling follicular growth, meiotic maturation, vascular remodeling of the ovary, and the synthesis of steroid hormones. Its discovery dates back to 1910, when Thomas Hunt Morgan first described mutations affecting this pathway. In mammals, four type I transmembrane receptors (NOTCH1–NOTCH4) interact with five type I transmembrane ligands (JAG1, JAG2, DLL1, DLL3, DLL4), thereby initiating Notch-dependent signaling cascades [67]. NOTCH1 is a high-molecular-weight transmembrane receptor, 300˜kDa, characterized by a complex architecture. Its extracellular domain contains multiple epidermal growth factor (EGF)-like repeats, alongside Lin12-type motifs and a region that is essential for heterodimer formation. This is succeeded by a solitary membrane-spanning segment. The intracellular portion of the protein comprises a transactivation domain and a PEST-rich sequence, which is notable for its abundance of proline, glutamate, serine, and threonine residues [68]. These residues contribute to protein turnover and regulatory control. Upon ligand engagement, the receptor undergoes conformational changes that facilitate interaction with CSL family transcription factors. This association has been demonstrated to promote the assembly of transcriptional activation complexes, which in turn drive the expression of Hairy and Enhancer of Split (HES) genes. These genes encode basic helix-loop-helix (bHLH) proteins that play pivotal roles in cell fate determination and differentiation processes (see Figure 5) [69]. Experimental studies in mice have revealed distinct patterns of Notch receptor expression in the ovary. NOTCH1 is located in the endothelial cells of the theca layer during follicle growth and later in the newly formed vessels of the corpus luteum, as well as in the mature vascular system of the theca, with expression persisting during pregnancy in hormonally stimulated ovaries. NOTCH4 is present in PECAM-positive endothelial cells during folliculogenesis and corpus luteum development, while NOTCH2 and NOTCH3 are mainly detected in granulosa cells of developing follicles [70]. Genetic or experimental disruption of the Notch pathway in mice has been shown to impair meiotic spindle organization, follicle morphogenesis, granulosa cell proliferation and survival, corpus luteum function, and ovarian neovascularization. This pathway is highly conserved and tightly regulated, and Notch signaling abnormalities are implicated in many human diseases, including hereditary syndromes and various cancers [71,72,73]. Importantly, the expression of Notch receptors, ligands, and regulatory proteins is dynamic during mammalian folliculogenesis, and signaling occurs between germ cells and granulosa cells, within the granulosa cell population, and between ovarian vascular endothelial cells.

### 3.4. The PI3K/Akt

The PI3K/Akt pathway is activated when growth factors, such as VEGF, FGF and IGF-1, bind to their respective receptors, including VEGFR2, FGFR and IGF1R [74]. This signaling cascade is essential for ovarian angiogenesis, ensuring the supply of nutrients and hormones to developing follicles. Following the interaction of the ligand and receptor, the receptor tyrosine kinases (RTKs) undergo dimerization and the phosphorylation of specific tyrosine residues, enabling the recruitment of signaling molecules such as PI3K [75]. In the ovary, VEGF is particularly important because it promotes endothelial cell proliferation and increases vascular permeability [76]. PI3K is composed of a catalytic (p110) and a regulatory (p85) subunit. Upon activation, PI3K phosphorylates phosphatidylinositol-4,5-bisphosphate (PIP2), generating phosphatidylinositol-3,4,5-trisphosphate (PIP3) at the plasma membrane. PIP3 has been identified as a docking site for proteins with pleckstrin homology (PH) domains, including Akt (protein kinase B, PKB) [77]. Akt, a serine/threonine kinase, is known to be activated through phosphorylation by PDK1 at Thr308 and by the mTORC2 complex at Ser473. Once activated, Akt phosphorylates multiple downstream effectors that regulate survival, proliferation, and migration of cells [78]. Akt, a serine threonine kinase, has been shown to promote cell survival and migration in endothelial cells of the ovary, thereby facilitating new vessel formation. Akt has been demonstrated to be a significant regulator of mTOR, the catalytic kinase of both mTORC1 and mTORC2. The mTORC1 complex is imperative for protein synthesis and angiogenesis, in part by stimulating the translation of VEGF and other proangiogenic mediators [79]. In the context of ovarian tissue, the Akt/mTOR axis has been demonstrated to reinforce angiogenesis by driving VEGF expression and signaling [80,81]. Akt activates endothelial nitric oxide synthase (eNOS), leading to nitric oxide (NO) production, which further promotes vasodilation, endothelial proliferation, and vascular permeability [79,82]. From a physiological standpoint, NO is imperative for the regulation of blood flow during the processes of ovulation and corpus luteum development. Dysregulated PI3K/Akt signaling has been linked to various ovarian pathologies, including PCOS and cancer. Its overactivation has been demonstrated to promote excessive angiogenesis, abnormal follicle growth, and tumor progression [83,84].

### 3.5. HIF Pathway

The hypoxia-inducible factor (HIF) signaling axis enables cells to adapt to hypoxic conditions and directly influences angiogenesis. HIF is a heterodimer consisting of an oxygen-sensitive α-subunit (the best-studied example being HIF-1α) and a constitutive β-subunit. Under normoxic conditions, prolyl hydroxylase domain proteins (PHDs) hydroxylate HIF-1α, promoting recognition by the von Hippel–Lindau protein (pVHL) and subsequent ubiquitin-mediated proteasomal degradation [85]. During hypoxia, however, HIF-1α escapes degradation, accumulates and translocates into the nucleus, where it dimerizes with HIF-1β to form an active transcriptional complex [86]. HIF-1 controls the transcription of numerous hypoxia-responsive genes, many of which directly regulate angiogenesis. One such gene is vascular endothelial growth factor (VEGF), which is the main mediator of endothelial cell proliferation, migration and differentiation, thereby enabling new vessel formation [87]. In addition to VEGF, HIF-1 stimulates the expression of angiopoietins (ANGPT1 and ANGPT2), stromal cell-derived factor-1 (SDF-1/CXCL12) and the receptor tyrosine kinase Tie2. These factors all contribute to vascular remodeling [88]. Furthermore, HIF-dependent signaling modifies endothelial metabolism and induces proteolytic enzymes, such as matrix metalloproteinases (MMPs), which degrade the extracellular matrix and allow endothelial cells to invade surrounding tissue, thereby promoting angiogenesis [89].

## 4. Signaling Pathways—Applications in Medicine

A comprehensive analysis of the role of angiogenesis signaling factors provides the foundation for research exploring their functions and signaling pathways in which they participate. The most importantly, the review on angiogenesis-regulating molecules points out their potential applications in medical therapies. Ovarian cancer is one of the most aggressive types of gynecological cancers, in which angiogenesis plays a pivotal role in its progression. In ovarian cancers, elevated expression of angiogenic factors facilitates the formation of new blood vessels that provide cancer cells with nutrients and oxygen, thereby promoting their rapid growth and metastasis. The objective of antiangiogenic therapies is to inhibit this process. A number of studies described the role of HIF-1 in cancer. These include immunohistochemical analyses of HIF-1 expression in tumor biopsies, which provided prognostic information and identified groups of patients requiring more aggressive therapy [90]. The disruption of the HIF-1 pathway through the use of dominant negative HIF-1 was demonstrated to be an effective treatment for pancreatic cancer. This approach was shown to reduce the tumorigenicity of pancreatic cancer cells by inhibiting glucose metabolism and by sensitizing cancer cells to hypoxia-induced apoptosis and growth inhibition [91]. The polypeptide HIF-1 C-TAD, which competes for binding with CBP/p300, was demonstrated to reduce the expression of VEGF and the growth of tumors in mice [92]. A number of novel therapeutic agents which target signaling pathways have been developed. Many of them block HIF-1 function and exert antiangiogenic activity. Examples of such agents include Calphostin C (protein kinase C inhibitor); trastuzumab (Herceptin); gefitinib (Iressa); wortmannin and LY294002 (PI3K inhibitors); and PD98095 (MAPK inhibitor). Rapamycin, an inhibitor of the rapamycin-related protein FKBP12 or mammalian target of rapamycin (FRAP/mTOR), was also identified as a potential agent. Similarly, diphenyleneiodonium (a redox signal blocker) and mannoheptulose (a glucokinase inhibitor) were proposed as possible candidates [93]. In PCOS, the development of ovarian cysts and ovulatory dysfunction are attributed to excessive angiogenesis and abnormalities in PDGF, VEGF, and MMP signaling. The use of therapeutic modalities that regulate angiogenesis may prove beneficial in the management of these pathological conditions. Elevated levels of VEGF in individuals with PCOS were linked to excessive angiogenesis, which may contribute to the formation of ovarian cysts. The potential use of pharmacological agents that reduce VEGF levels or block its action in patients with PCOS is a promising avenue of research as a means of controlling angiogenesis and improving ovarian function. A deeper comprehension of angiogenesis signaling pathways, including the PI3K/Akt pathway and the TGF-β pathway, offers promising avenues for novel therapeutic interventions in the management of both cancer and reproductive disorders. The majority of studies are concentrated on the identification of novel angiogenesis inhibitors that may prove effective in the treatment of refractory forms of ovarian cancer. The other approach is focused on the stimulation of angiogenesis with a view to enhance the efficacy of reproductive therapies. The PI3K/Akt pathway plays a pivotal role in the proliferation and migration of endothelial and tumor cells. Inhibitors of this pathway, such as wortmannin and LY294002, are currently being investigated as potential anti-angiogenic agents in ovarian cancer. The inhibition of PI3K activity may serve to restrict tumor angiogenesis, thereby impeding the capacity of the tumor to grow and metastasize [50]. Further investigations into the cross-talk between various angiogenic pathways may yield new therapeutic targets with enhanced specificity and efficacy. It is imperative to understand the temporal and spatial expression patterns of angiogenic factors in both normal and pathological ovarian tissue in order to develop targeted interventions. While this review primarily focused on downstream signaling and angiogenic outcomes, it is important to emphasize that the upstream regulation of these pathways—by hypoxia, endocrine, and inflammatory signals—remains an active and rapidly evolving research area that warrants further dedicated reviews. Furthermore, the development of novel methodologies for the identification of biomarkers has the potential to expedite the process of diagnosis and the implementation of personalized treatment strategies. The combination of antiangiogenic therapy with conventional chemotherapeutic regimens has the potential to yield synergistic effects, thereby enhancing patient outcomes. A comprehensive understanding of angiogenesis in ovarian physiology and pathology will pave the way for the development of more precise and effective treatment modalities.

## Figures and Tables

**Figure 1 cells-14-01555-f001:**
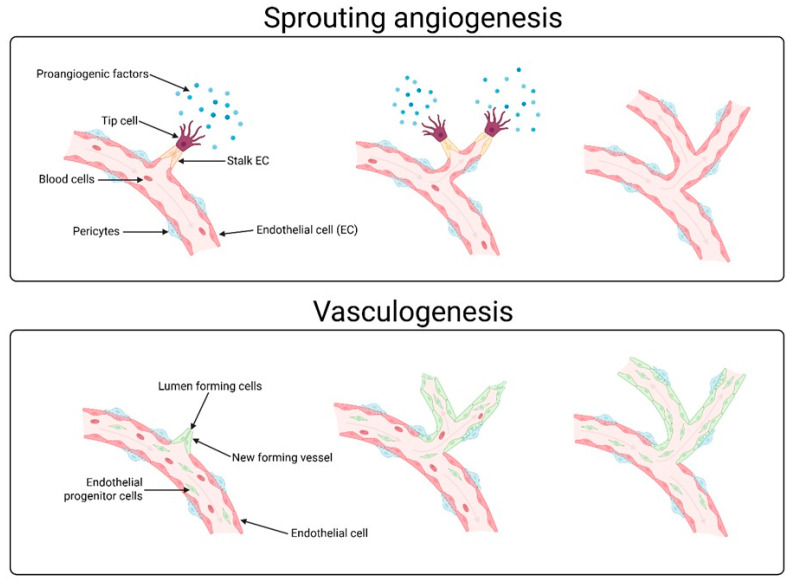
Graphical representation of sprout angiogenesis and vasculogenesis, and the differences between them. Angiogenesis is the process by which new blood vessels form from existing ones. This process involves tip cells and proangiogenic factors, among other things. Vasculogenesis, on the other hand, describes the creation of entirely new blood vessels directly from endothelial cells.

**Figure 2 cells-14-01555-f002:**
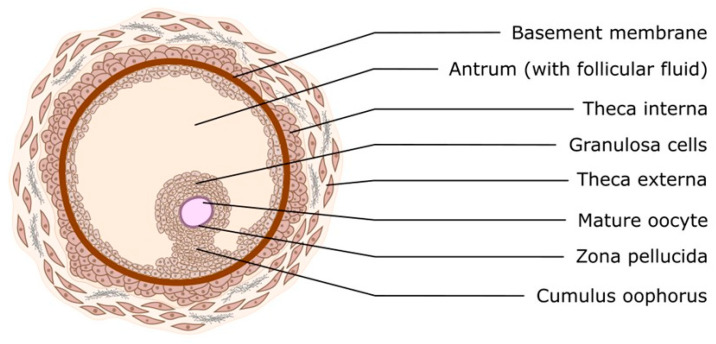
Morphology of ovarian follicle with mature oocyte.

**Figure 3 cells-14-01555-f003:**
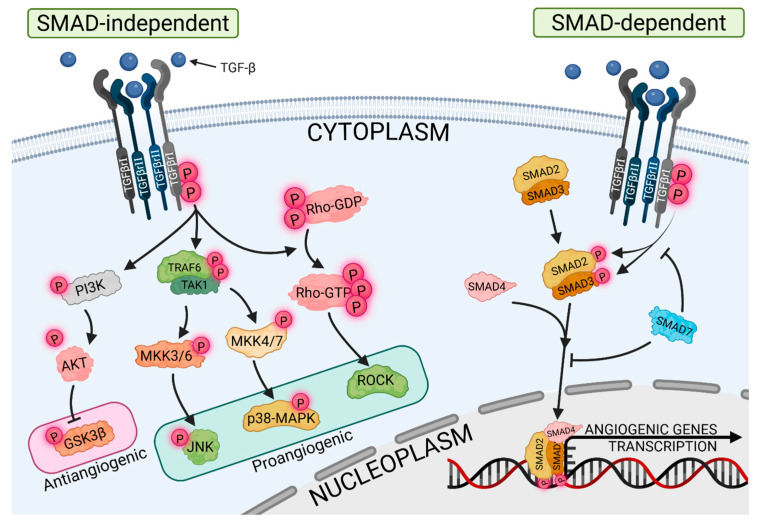
SMAD-independent and SMAD-independent angiogenic pathways of TGF-β (abb: TGF-β, transforming growth factor β; TGF-βI and TGF-β II, transforming growth factor β receptor type I and transforming growth factor β receptor type II; PI3K, Phosphotylinosital 3 kinase; TRAF6, TNF Receptor Associated Factor 6; TAK1, TGF-β-activated kinase 1; Rho-GDP, Rhodopsin-Guanosine Diphosphate; Rho-GTP, Rhodopsin-Guanosine Triphosphate; AKT, Protein kinase B; MKK3/4/6/7, mitogen-activated protein kinase kinase 3/4/6/7; GSK3β, Glycogen synthase kinase-3 beta; JNK, Jun N-terminal kinase; p38-MAPK, p38 Mitogen Activated Protein Kinase; ROCK, Rho-associated coiled-coil kinase).

**Figure 4 cells-14-01555-f004:**
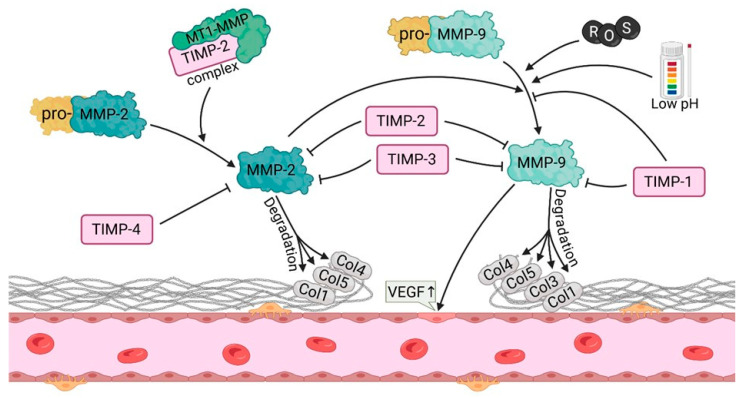
MMPs way of action, activation and regulation though TIMP in angiogenesis (abb: mmp-2/9, matrix metalloproteinase-2/9; MT1-MMP, membrane type 1 matrix metalloproteinase; TIMP1/2/3/4, tissue inhibitor of metalloprotease-1/2/3/4; ROS, reactive oxygen species; Col1/3/4/5, collagen type 1/3/4/5; VEGF, vascular endothelial growth factor).

**Figure 5 cells-14-01555-f005:**
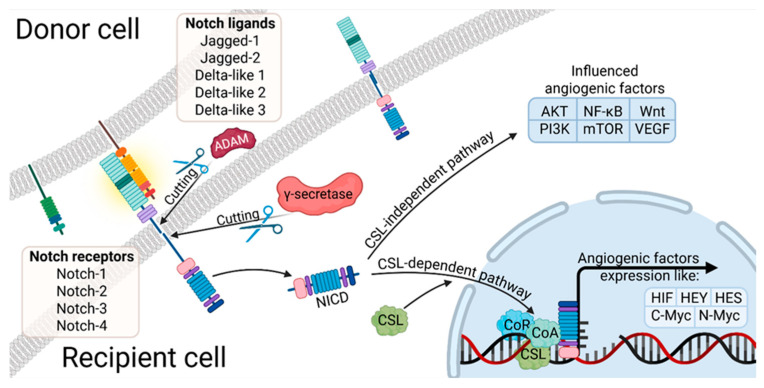
Way of Notch receptors way of action in CSL-independent and CSL-dependent pathways in angiogenesis (abb. ADAM, a disintegrin and metalloproteinase; NCID, Notch intracellular domain; AKT, protein kinase B; NF-B, nuclear factor kappa-light-chain-enhancer of activated B cells; Wnt, Wingless-related integration site; PI3K, phosphatidylinositol 3-kinase; mTOR, mammalian target of rapamycin; VEGF, vascular endothelial growth factor; CSL, CBF1, Suppressor of Hairless, Lag-1; CoR, Cold regulated genes; CoA, coagulase gene; HIF, Hypoxia-inducible factor 1; HEY, Hes Related Family BHLH Transcription Factor With YRPW Motif 1; HES, hairy and Enhancer of split).

## Data Availability

No new data were created or analyzed in this study.

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
