# Peer review of "The Role of Growth Factors and Signaling Pathways in Ovarian Angiogenesis"

_cells, 2025, doi:10.3390/cells14191555_

Round 1
Reviewer 1 Report
Comments and Suggestions for Authors
This review is a great topic to ovarian researcher. Ovarian angiogenesis involves a coordinated network of signaling pathways and molecular factors. Vascular dysfunction and impaired angiogenesis are key contributors to various ovarian disorders and infertility. Therefore, in-depth studies of ovarian vascularization are crucial for identifying the pathophysiology of these conditions and guiding the development of effective treatments. The review is detailed, comprehensive, and has a certain depth. However, this paper should be revised carefully before being considered to be accepted to CELLS.
- The abbreviations in the entire paper are very non-standard, with repeated abbreviations, abbreviations written directly without the full name, and abbreviations not written in full and in alphabetical order at the end of the text.
Such as:
(1)P2 Line 29-30 Vascular endothelial growth factor (VEGF), fibroblast growth factor (FGF), prostaglandin F2; PGF)
(2)Transforming Growth Factor-beta (TGF-)
(3)a range of processes associated with cell proliferation, migration, and
differentiation. Its receptor, MET, i
(4)Abbreviations
The following abbreviations are used in this manuscript:
VEGF Vascular endothelial growth factor
HIF hypoxia-inducible factor
TGF- Transforming Growth Factor-beta
HGF Hepatocyte Growth Factor
PDGF Platelet-Derived Growth Factor
MMPs Matrix Metalloproteinases
- P3 Line 78-124 HIF (hypoxia-inducible factor)and P12 Line 430-451 HIF pathway.
In terms of content, there are many repetitions between the two.
- The reasons for the generation of these angiogenic factors (eg. VEGF, FGF, PDGF, TGF, et al.)are not covered in many reviews, which is also of interest to readers. If this part of the content can be discussed, it will make the review more in-depth.
- The example of vascular dysfunction and PCOS is somewhat inappropriate, as the mechanism of PCOS is mainly related to androgen imbalance.
- Figure 1. Graphical representation of sprouting angiogenesis, vasculogenesis and their differences
In the caption, it is necessary to provide a detailed explanation of these two processes. Just drawing the diagram may confuse the reader.
- There are still a large number of spelling errors in the article, and the entire text needs to be carefully revised.
For example:
P5 Line 122 angiogenesis-relatedgenes
, it can participate inthe repair of injury
Transforming Growth Factor-beta (TGF-)
Author Response
We thank the Reviewer for the thoughtful and constructive comments. Please see the attachment.

Reviewer 2 Report
Comments and Suggestions for Authors
In this review article, the authors summarized angiogenic factors on ovarian angiogenesis and discussed a possible therapeutic approatch on the ovarian diseases such as ovarian cancer and PCOS. The team are important and interesting, but the article appears to have several problems which should be corrected.
Comments.
1, Most of the cited articles appear to be published more than 10-15 years ago, thus, the recent progress in many research fields seems not well introduced.
2, About HIF: in the introduction, the authors did not clearly describe that the HIF is a transcription factor, but not a general growth factor such as VEGF and FGF. Thus, the authors should describe it more clearly. Furthermore, it is well known that the HIF works as a heterodimer: HIFa (alpha) and HIFb (beta), and the major factor of HIFa family is HIF1a, but HIF2a also plays important role in several tissues such as placental trophoblasts. The authors should describe the naming more clearly, and also should check more carefully which members of HIF are the important player in physiological and pathological conditions in ovary.
3, line 53: “showed” should be “shown”? “figure 1” should be “figure 2”?
4, Page 9, VEGFR: many amino acid numbers appear to be wrong in this section; for example, “mature protein (201356 aas)” should be corrected.
5, The authors described several possible antiangiogenic materials, chemicals and antibodies, for the treatment of ovarian diseases. They should check more clearly the recent progress: how were the results of clinical trials on the treatment of such ovarian diseases, and the results after the approval for clinical use. Also, the authors described a little on the dominant negative (dn) form of HIF. Is such a dn-HIF well developed for suppression of the ovarian diseases? If not, what is the problem?
Comments on the Quality of English Language
English language in the whole article should be carefully checked and corrected.
Author Response

(The authors gave the same response as above.)

Round 2
Reviewer 1 Report
Comments and Suggestions for Authors
The author has responded to my comments, but there are still a large number of spelling and abbreviation errors throughout the article. For example:
- P5 Line 160: proteins are often present at undetectable levels.
- P6 Line 213: TGF-β signalling can have pathological implications. In ovarian cancer, for example, TGF-betait
- P6 Line 219: Figure 3. Figure 3
; TGF-βI & II, transforming growth factor β receptor I & II;
- P6 Line 227-246:Hepatocyte Growth Factor (HGF)
A multifunctional cytokine that plays an important role in regulating a number of processes in the context of angiogenesis, i.e. those related to proliferation, cell migration and differentiation, is hepatocyte growth factor (HGF).
- P11 Line 442: molecular-weight transmembrane receptor (300˜kDa) characterised by a complex archi-
The author has responded to my comments, but there are still a large number of spelling and abbreviation errors throughout the article. For example:
- P5 Line 160: proteins are often present at undetectable levels.
- P6 Line 213: TGF-β signalling can have pathological implications. In ovarian cancer, for example, TGF-betait
- P6 Line 219: Figure 3. Figure 3
; TGF-βI & II, transforming growth factor β receptor I & II;
- P6 Line 227-246:Hepatocyte Growth Factor (HGF)
A multifunctional cytokine that plays an important role in regulating a number of processes in the context of angiogenesis, i.e. those related to proliferation, cell migration and differentiation, is hepatocyte growth factor (HGF).
- P11 Line 442: molecular-weight transmembrane receptor (300˜kDa) characterised by a complex archi-
Author Response
Dear Reviewer,
Thank you very much for your careful reading of our manuscript and for providing valuable comments. All the errors and unclear expressions you identified have been corrected and are highlighted in the text using the track-changes mode for your convenience.
In addition, the entire manuscript has undergone thorough language editing by a native English speaker to further improve grammar, vocabulary, and overall clarity.
We hope that these revisions meet your expectations and significantly enhance the quality of our work.
Sincerely,
Hanna Jankowska-Ziemak
On behalf of all authors
Reviewer 2 Report
Comments and Suggestions for Authors
The authors responded and revised most of the comments from the reviewer.
Author Response
Dear Reviewer,
Thank you very much for your careful reading of our manuscript and your valuable comments, which undoubtedly improved the quality of the review. The entire manuscript has been thoroughly edited by a native English speaker, further improving grammar, vocabulary, and overall clarity.
We hope that these revisions will meet your expectations and significantly improve the quality of our work.
Sincerely,
Hanna Jankowska-Ziemak
On behalf of all the authors
Round 3
Reviewer 1 Report
Comments and Suggestions for Authors
NO